# Removal of Cu(II) Contamination from Aqueous Solution by Ethylenediamine@β-Zeolite Composite

**DOI:** 10.3390/molecules26040978

**Published:** 2021-02-12

**Authors:** Peng Liu, Hui Ruan, Tiantian Li, Jiaqi Chen, Fuqiu Ma, Duoqiang Pan, Wangsuo Wu

**Affiliations:** 1College of Nuclear Science and Technology, Harbin Engineering University, Harbin 150001, China; liup15@lzu.edu.cn (P.L.); huiruan@hrbeu.edu.cn (H.R.); litt@hrbeu.edu.cn (T.L.); chenjiaqi@hebeu.edu.cn (J.C.); 2Yantai Research Institute and Graduate School, Harbin Engineering University, Yantai 264006, China; 3Radiochemistry and Nuclear Environment Laboratory, School of Nuclear Science and Technology, Lanzhou University, Lanzhou 730000, China; wws@lzu.edu.cn; 4Key Laboratory of Special Function Materials and Structure Design, Ministry of Education, Lanzhou 730000, China

**Keywords:** adsorption, Cu(II) contamination, Ethylenediamine, β-zeolite

## Abstract

The low cost β-zeolite and ethylenediamine modified β-zeolite (EDA@β-zeolite) were prepared by self-assembly method and used for Cu(II) removal from contaminated aqueous solution. Removal ability of β-zeolite toward Cu(II) was greatly improved after ethylenediamine (EDA) modification, the removal performance was greatly affected by environmental conditions. XPS results illustrated that the amide group played important role in the removal process by forming complexes with Cu(II). The EDA@β-zeolite showed desirable recycling ability. The finding herein suggested that the proposed composite is a promising and suitable candidate for the removal of Cu(II) from contaminated natural wastewater and aquifer.

## 1. Introduction

Copper is one of the most common pollutants in water environment around industrial sites, such as paint, metal finishing, electroplating, mining operations, and fertilizer [1]. Excessive uptake of copper in biosphere would accumulate in organism through the water cycle, and cause abdominal pain, vomiting, even movement and neurological disorder [2,3,4,5]. Therefore, the efficient removal of Cu(II) from contaminated aqueous solutions is crucial to mitigate the hazard of copper toward environment. Among many purification technologies, the sorption method has drawn interest because of its high efficiency, easy operation, and low cost. Many adsorbents, such as oxides [6], porous material [7], chelating resin [8], etc., have been widely proposed for the purification and remediation of contaminated water bodies. However, the shortages such as high cost, tedious synthesis process, undesirable removal performance, and environmental toxicity, restricted the further industrial applications. Therefore, an economical and environment friendly sorbent with prominent removal performance need to be developed for the removal of copper contaminants from aqueous solutions.

β-zeolites have a unique three dimensional structure that constituted with a 12-membered rings which is made up of four five-membered rings consists of SiO_4_ and AlO_4_ tetrahedra [9], the 12 atoms structure consists of two types of intersecting channels, which makes β-zeolite different from other zeolites. The neat porous structure of β-zeolites makes it attracting considerable attentions in the field of catalysis, separation, and water treatment [10,11]. However, the removal capacity of raw zeolite toward heavy metal ions is usually limited due to the low reactivity of componential elements of zeolite. Therefore, functional groups with specific coordination ability are widely considered to integrate onto raw zeolites to enhance the removal capacity and selectivity toward heavy metals. Panneerselvam et al. [12] found that the phosphoric acid modification on β-zeolite enhanced the removal of copper ion from aqueous solution. The decoration on zeolite with hydrated aluminum oxide, manganese oxide, aminopropyltriethoxysilane, etc., displayed considerable improvement on copper removal performance [13,14]. Many inorganic and organic materials, such as carbon materials, biosorbents, proteins, and so on, have been used to adsorb metal ions, but for applications under commercially relevant operating conditions, a type of thermal stable and inexpensive microporous materials is much needed. Therefore, more promising zeolite-based composites with convincing merits are demanded in contaminated effluents treatment industry.

Ethylenediamine, which is one of the most typical widely distributed chelating ligand, displays outstanding chelating ability to heavy metal ions because of two nitrogen atoms donating lone pairs of electrons. As thus, one amine ligand of ethylenediamine is used as organic linkers to coordinate zeolite, the other amine ligand is applied to adsorb metal ions. Therefore, ethylenediamine and zeolite, with low-cost, high-efficiency, and easy-availability merits, were easily considered to be integrated as a promising composite. This hybrid combines the advantages of porous network for zeolite and high cation affinity for ethylenediamine. In present work, ethylenediamine grafted β-zeolite (EDA@β-zeolite) was prepared and used for Cu(II) removal from aqueous solution; the removal performance was examined by batch removal test under various environmental conditions; the related removal mechanism was discussed with the aid of spectroscopy, such as X-ray powder diffraction (XRD), X-ray photoelectron spectroscopy (XPS), etc. This composite is a potential material in the successful decontamination of copper containing waste water.

## 2. Experimental

### 2.1. Materials

Copper chlorite, tetraethyl ammonium hydroxide (TEAOH, 20 wt% in water) and ethylenediamine were purchased from Aladdin Reagent Co. Ltd. β-zeolite was synthesized by hydrothermal method, as reported in previous work [10,15]. The obtained solid product was filtered and washed with deionized water. After drying at 96 °C overnight, the solid was calcined at 540 °C for 20 h. The one-pot synthesis of ethylenediamine functionalized β-zeolite followed the procedure described in our previous work. Briefly, 1.5 g of β-zeolite was mixed with 30 mL of EDA solution, the mixture was refluxed at 120 °C for 40 h. Then the solid was filtered and washed with a 0.1 M NaCl solution until no amine was detected. NaCl remaining in the sample was further washed with water and methanol. The solid were dried at 70 °C and designated as EDA@β-zeolite [15,16]. All other chemicals used were analytical grade and used without further purification. All solutions and suspensions were prepared with deionized water (18 MΩ·cm^−1^).

In order to verify the stability of EDA@β-zeolite, leaching experiments were carried out. A certain amount of EDA@β-zeolite and deionized waters were added into a series of polyethylene centrifuge tubes, pH of the suspensions was adjusted to 4–12. After 72 h, the suspensions were centrifuged, the solid were dried and weighed. The leaching efficiency was calculated from the difference between initial weight and the weight after drying: leaching efficiency % = (*R*_0_ − *R*)/*R*_0_ × 100%, where *R*_0_ (g) is the initial weight of EDA@β-zeolite, *R* (g) is the weight of EDA@β-zeolite after drying. The results (Appendix A) were revealed that EDA@β-zeolite has good stability in pH range from 4 to 12.

### 2.2. Characterization

The Fourier transform infrared spectroscopy was recorded by potassium bromide pellet (NEXUS 670, Nicolet). X-ray powder diffraction (PANalytical X’Pert PRO) patterns were collected using a Cu-Kα radiation source (λ = 0.154 nm) in the range of 5 ≤ 2θ ≤ 60° with a step size of 0.05°. The morphological images of β-zeolite and EDA@β-zeolite were obtained on transmission electron microscope (Hitachi Model H-600) and scanning electron microscopy (Hitachi S-4800). X-ray powder diffraction (PANalytical X’Pert PRO) patterns were collected using a Cu-K_α_ radiation source (λ = 0.154 nm) in the range of 5 ≤ 2θ ≤ 60° with a step size of 0.05°. The samples for XPS measurement were prepared in proportional amplified condition by exactly following the procedures in removal experiment. The wet paste after centrifugation were collected and dried at 40 °C under N_2_ conditions before the X-ray photoelectron spectroscopy (Thermo ESCALab 220i-XL) determination. The XPS data were collected at 300 W with Mg K_α_ radiation, and the binding energies were corrected using C 1 s peak at 284.80 eV as a reference.

### 2.3. Batch Experiments

A certain amount of β-zeolite or EDA@β-zeolite were spiked into a series of polyethylene centrifuge tubes, Cu(II) solution, NaCl and deionized water solution were added, the total volume is 6.0 mL. The small volume of NaOH or HCl solutions were added to adjust pH, then the tubes were placed into a thermostatic oscillator for 48 h. After reaching equilibrium, the tubes were centrifuged at 10,000 r/min for 30 min to separate the solid and liquid phase. A certain volume of supernatant was piped out to measure the aqueous Cu(II) concentration by bis-cyclohexanone oxalyldihydrazone using spectrophotometer at a wavelength of 540 nm. The adsorption percentage of Cu(II) was calculated from the difference between initial and equilibrium concentrations (Adsorption % = (*C*_0_ − *C_eq_*)/*C*_0_ × 100%, and *C_s_* = (*C*_0_ − *C_eq_*) · *V/m*, where *C*_0_ (mol/L) is the initial concentration of Cu(II), *C_eq_* (mol/L) is the measured equilibrium concentration of Cu(II) in supernatant, *C_s_* (mol/g) is the amount of adsorbed Cu(II), *V* (L) and *m* (g) is the volume of suspension and the mass of adsorbent, respectively).

The regeneration of exhausted adsorbents was performed in 0.05 mol/L HCl solution. After being shaken for 24 h, the solid phase was separated and washed with deionized water, then the adsorbents were dried and reused for Cu(II) removal by following the procedures described above.

To inspect the adsorptions of Ca(II), Fe(III), and Cu(II) onto the β-zeolite and EDA@β-zeolite in mixed multi-metal solution, the batch adsorption experiments of effect of coexisting ions were done, in which CaCl_2_, FeCl_3_ and CuCl_2_ were applied to obtain a mixed multi-aqueous solution. During this process, NaOH and HCl were used to adjust pH. After 48 h, the concentration of Ca(II), Fe(III) and Cu(II) were measured by Inductively Coupled Plasma-Atomic Emission Spectrometry (ICP-AES).

## 3. Results and Discussion

### 3.1. Effect of pH on Cu(II) Removal

The effect of pH on Cu(II) removal by β-zeolite and EDA@β-zeolite is shown in Figure 1; the results show that Cu(II) removal is strongly dependent on pH. At pH < 7.0, the removal of Cu(II) onto both β-zeolite and EDA@β-zeolite increased with pH increasing, the removal percentage by EDA@β-zeolite was obviously higher than that by β-zeolite under the same condition. The experimental phenomenon of the prepared materials is similar to many other results of metal ions adsorption from references [16,17,18]. It was conceivable that an excess of positively charged hydrogen ions existed in system at low pH, which caused that the competition between both positively charged Cu(II) and hydrogen ions for the available sorption sites on β-zeolite and EDA@β-zeolite surface. As pH increased, the competition between Cu(II) and hydrogen ions decreased, the electrostatic repulsion between Cu(II) and adsorbents surface decreased as well, which resulted in the sharp rise of Cu(II) removal in the pH range of 4.0–7.0 [16,19,20,21]. When pH above 7.0, the removal of Cu(II) reached maximum plateau and changed inconspicuously.

### 3.2. Removal Kinetics

The influence of contact time on Cu(II) removal was shown in Figure 2. Due to the formation of insoluble substance in high pH, the influence of contact time was explored at pH = 5.1 and 6.1. The removal percentage of Cu(II) increased with contact time increasing until removal achieved maximum at ~20 h, indicating that the removal of Cu(II) by both β-zeolite and EDA@β-zeolite were kinetically fast. The removal percentage at pH 6.1 was higher than that at pH 5.1, this result further confirmed the previous pH-dependent results. The kinetic information on removal process could be used not only for predicting the rate of Cu(II) removal, but also for elucidating the underlying removal mechanism. Therefore, three kinetic models were employed to understand the kinetic processes of Cu(II) removal on β-zeolite and EDA@β-zeolite.

Firstly, the pseudo-first-order equation was employed, it could be expressed as following [22]:(1)$$\frac{1}{q_{t}}=\frac{K_{1}}{q_{e}}\times \frac{1}{t}+\frac{1}{q_{e}}$$
where *q_t_* (mg·g^−1^) and *q_e_* (mg·g^−1^) represent the amounts of adsorbed Cu(II) at the time *t* (h) and equilibrium, respectively, *K*_1_ (h^−1^) is the first-order constant of adsorption rate. The corresponding fitting results were listed in Table 1. The experimental *q_e_* values did not match the theoretical *q_e1_* values, suggesting that the pseudo-first-order model was not applicable to predict the Cu(II) adsorption systems [23].

Basing on pseudo-first-order model, the pseudo-second order equation was further analyzed for the sorption kinetics [22]. The pseudo-second-order kinetic linear expression can be presented as following:(2)$$\frac{t}{q_{t}}=\frac{1}{K_{2}q_{e}^{2}}+\frac{t}{q_{e}}$$
where *q_e_* (mg·g^−1^) is the equilibrium adsorption amount, *q_t_* (mg·g^−1^) is the adsorption amount at time *t* (h). The parameter *K*_2_ (g·mg^−1^h^−1^)) represents the second-order constant of adsorption rate. From Table 1, one can see that the *R_2_*^2^ were much closer to unity than *R*_1_^2^, which elucidated that the pseudo-second-order model described the removal processes better (Figure 2B), indicating that the chemical adsorption rather than physical adsorption was the main mechanism for Cu(II) removal [24,25].

The Weber–Morris model is an empirically functional relation and is widely used to describe the sorption process in porous media. The Weber–Morris model can be described as following:(3)$$q_{t}{\;=\;Kt\;}^{\frac{1}{2}}\;+\;C$$

According to Equation (3), the plot of *q_t_* vs. *t*^1/2^ was linear, where *K* is the slope and *C* is the intercept. The multi-linear plots in Figure 2C indicated that multiple mechanisms controlled the Cu(II) removal process [26]. The removal process was partitioned into three stages: initial stage, second stage, and equilibrium stage. The initial plots were a steep slope, this stage could be attributed to the instantaneous adsorption or the external surface adsorption on the most available surface sites on β-zeolite or EDA@β-zeolite. The second stage was a gentle slope, which was ascribed to the gradual adsorption. The third stage was the equilibrium stage, the plots were nearly horizontal lines. These results revealed that the intra-particle diffusion was the mainly rate-controlled step. At the beginning of the reaction, Cu(II) was adsorbed by the exterior surface, after adsorption sites of exterior surface reaching saturation, Cu(II) diffused into adsorbents along the pores and the adsorption process on interior surface started. With Cu(II) diffusing into intra-pores, the diffusion resistance increased, which caused the decrease in diffusion rate until the diffusion process reached equilibrium. The fitting parameters for Weber–Morris model were listed in Table 2, the rate constants (*K*_1_, *K*_2_, *K*_3_) were corresponded with the three adsorption stages of exterior surface stage, interior surface stage and equilibrium stage, respectively. Note that at the same stage, the rate constants of β-zeolite were higher than that of EDA@β-zeolite before equilibrium, indicating that the micro-pores on β-zeolite became smaller or partially blocked after ethylenediamine modification, therefore the diffusion and transport of Cu(II) into interior pores of EDA@β-zeolite became more difficult. The increased values of *C* (μg/g) as time increasing indicated that the Cu(II) adsorption process became less influenced by the thickness of boundary layer [27].

### 3.3. Thermodynamic Estimation

The adsorption isotherms of Cu(II) on β-zeolite and EDA@β-zeolite at different temperatures were shown in Figure 3. The removal of Cu(II) was strongly dependent on temperature, high temperature was favorable for Cu(II) removal. The thermodynamic parameters (Δ*H*^0^, Δ*S^0^,* and Δ*G*^0^) can be calculated from these temperature-dependent adsorption isotherms. The Gibbs free energy change (Δ*G*^0^) can be calculated from:*ΔG*^0^ = *−RTlnK*^0^(4)
where *R* represents the ideal gas constant (8.314 J^.^mol^−1.^K^−1^), *T* (K) is the temperature in Kelvin, and *K*^0^ is the thermodynamic equilibrium constant. The equilibrium partition coefficient (*K_d_*) can be presented as:*K_d_* = *q_e_*/*C_e_*(5)
where *q_e_* (mol·g^−1^) is the concentration of adsorbed Cu(II), *C_e_* (mol·L^−1^) is the Cu(II) concentration in aqueous. The value of ln*K*^0^ can be extrapolated by plotting ln*K_d_* against *C_e_* when *C_e_* is close to zero [28]. The standard entropy changes (Δ*S*^0^) and the average standard enthalpy changes (Δ*H*^0^) can be expressed basing on the following equations:(6)$${\Delta S}^{0}=\;-{\left(\frac{\;\partial {\Delta G}^{0}}{\partial T}\right)}_{p}$$

Δ*H*^0^ = Δ*G*^0^ + *T*Δ*S*^0^(7)

The corresponding thermodynamic parameters were listed in Table 3, the negative Δ*G*^0^ values reflected the adsorption process of Cu(II) on β-zeolite and EDA@β-zeolite were spontaneous. With temperature increasing, the values of Δ*G*^0^ decreased, implying the adsorption process had greater driving force at higher temperature. The positive Δ*H*^0^ values indicated that the overall adsorption process was endothermic. The entropy change (Δ*S*^0^) reflected the affinity of Cu(II) to β-zeolite or EDA@β-zeolite, the positive Δ*S*^0^ values were primarily ascribed to the release of hydration waters from hydrated Cu(II) ion before coordination with surface group [29].

### 3.4. Removal Isotherms

In order to understand the removal process better, Langmuir, Freundlich and Dubini–Radushkevich (D–R) models were applied to simulate the adsorption isotherms data, the corresponding fitting parameters were listed in Appendix A. Langmuir isotherm is usually valid for the monolayer sorption which contains a limited number of identical sites. The Langmuir isotherm could be described as the following form [30]:(8)$$\frac{C_{e}}{q}=\frac{{1\;+\;C}_{e}{\cdot K}_{a}}{K_{a}{\cdot q}_{\max}}$$
where *q_max_* (mg·g^−1^) is the adsorption capacity at saturation, *K_a_* (L·mmol^−1^) is the adsorption coefficient.

The Freundlich expression is an empirical equation, which allows for several different sorption sites on the adsorbents, and assumes that the adsorption occurs on a heterogeneous surface [31]. The linear equation could be presented as the following equation:*lnq* = *lnK_F_* + *n lnC_e_*(9)
where *K_F_* (mmol·g^−1^) and *n* are the Freundlich empirical constants, which represent the adsorption capacity and adsorption intensity, respectively. If *n* is between 1 and 10, the reaction is a favorable adsorption.

D–R isotherm model is more ordinary, because it assumes neither a constant adsorption potential nor a homogeneous surface. The D–R equation can be presented as [32]:(10)$$lnq_{m}=lnC_{s}\;+\;KR^{2}T^{2}{ln}^{2}\left(\frac{{1+C}_{e}}{C_{e}}\right)$$
where *K* is the constant related to adsorption energy, *q_m_* (mol/g) is the theoretical saturation capacity; *R* (kJ·mol^−1^K^−1^) is the gas constant and *T* (K) is the temperature in Kelvin. The free energy *E* (kJ·mol^−1^) is usually used for estimating the type of adsorption reaction, which could be described by the following equation:(11)$$E\;=\sqrt{\frac{1}{2K}}$$

If *E* < 8 kJ·mol^−1^, it could be supposed that the adsorption is governed by physical forces; if the value of *E* is between 8 and16 kJ·mol^−1^, chemical ion-exchange is the main adsorption mechanism; and if *E* is above 16 kJ·mol^−1^, the adsorption is affected by the particle diffusion process [33].

From the fitting parameters listed in Table 4, the relevant *R*^2^ indicated that the Langmuir model fitted the experimental data better. This fact indicated that the surface sites of β-zeolite and EDA@β-zeolite had uniform activity for Cu(II) adsorption, the adsorption was monolayer, and the adsorbed copper did not have further interaction with aqueous copper species. For Freundlich model, *n* value (between 1 and 10) showed the interaction intensity between Cu(II) and adsorbents, the increase in *n* value with temperature growing indicated the growth of adsorption heterogeneity. The *E* values obtained from D–R model were above 16 kJ·mol^−1^, suggesting that diffusion process played an important role in Cu(II) adsorption, which fell in right line with the kinetic results. Note that the adsorption capacities *q_m_* from the D–R model were quite different from the adsorption capacity *q_max_* values from Langmuir, this might be attributed to the different applicable assumption of each model [15,34,35,36,37].

### 3.5. Removal Mechanism

The Fourier transform infrared spectroscopy of β-zeolite and EDA@β-zeolite were showed Figure 4. The wide absorption band at 3100–3600 cm^−1^ was attributed to NH_2_, the bending vibration of H-O-H occurs at 1600–1645 cm^−1^, the absorption band at 1080–1090 cm^−1^ was the Si-O stretching vibration of Si-O-Si structure, the band at 792–795 cm^−1^ was attributed to the N-H bending vibration, and the bending vibration of 1485–1579 cm^−1^ in Figure 4b was corresponding to the symmetric bending vibration of NH_2_. A significant shift from 1088 to 1050 cm^−1^ after modification was observed, which was derived from the deceasing frequency of Si-O stretching due to the attachment of ethylenediamine.

Figure 5a shows the thermogravimetric curve of β-zeolite. The primary weight loss can be observed at about 110 °C, which is attributed to the desorption of water molecules. While three weight loss temperature can be observed at 100, 280, and 520 °C from Figure 5b, the additional weight loss temperatures correspond to the volatilization and degradation of EDA. The results showed that EDA was successfully combined with β-zeolite after the modification process [15].

The TEM images under different magnification were used to observe the changes in morphological features of β-zeolite before and after the modification of EDA. TEM image in Figure 6A–F display that β-zeolite and EDA@β-zeolite have smooth surfaces and exhibit a granular texture, there was no obvious morphological change. TEM images of β-zeolite and EDA@β-zeolite after adsorption were shown in Figure 6G,H,J,K, respectively, on which many black dots can be found, confirming the adsorption of Cu(II). Figure 6I,L were energy dispersive X-ray (EDX) elemental mapping of Cu. The full coverage of Cu revealed the adsorption performance of the adsorbents.

In order to examine the morphology of β-zeolite and EDA@β-zeolite, SEM is used to provide information on the morphological change before and after modification, meanwhile, the SEM coupled with energy dispersive X-ray spectroscopy (EDS) was used to assess the element distribution. The SEM images of β-zeolite and EDA@β-zeolite showed in Figure 7A,B. It can be clearly seen that β-zeolite and EDA@β-zeolite were aggregated spherical shape and diameter of particles that are in the range of about 0.3–0.5 μm. EDS spectra at single regions of β-zeolite and EDA@β-zeolite indicated the relative proportion of the different atoms, the results presented in Figure 7C,D, respectively. It is worth noting that the N mass percentage of β-zeolite and EDA@β-zeolite was about 0 and 54.74%, respectively. This result confirms the modification of EDA successfully.

X-ray diffraction patterns of β-zeolite and EDA@β-zeolite before and after adsorption were shown in Figure 8. In the scanning range from 5° to 60°, the characteristic peaks at 7.7° and 22.7° indicated that the fundamental structure of β-zeolite was preserved well after the EDA modification, even after Cu(II) adsorption the basic crystal structure of β-zeolite is remained. Nevertheless, a new peak at 29.3° appears after Cu(II) adsorption, it was assigned to CuCl_2_. Furthermore, the intensity of diffraction lines of β-zeolite and EDA@β-zeolite decreased after adsorption, all of these is due to the coverage of Cu(II) on the surface of adsorbents, the removal of Cu(II) is due to the adsorption of Cu(II) on β-zeolite and EDA@β-zeolite [10,15,38].

The XPS spectra for β-zeolite and EDA@β-zeolite before and after Cu(II) adsorption were showed in Figure 9. The XPS spectra of O1s for β-zeolite showed significant changes before and after Cu(II) adsorption (Figure 9A), the peaks at 532.10, 532.55, and 533.01 eV of oxygen shifted to higher binding energy after Cu(II) adsorption. This phenomenon was attributed to the decrease in electronic density of O atoms on β-zeolite because of the formation of complexes with Cu(II). Comparing the spectra of O 1s and N 1s of EDA@β-zeolite before and after Cu(II) sorption (Figure 9B,C), the peaks at 531.54, 532.06, and 532.58 eV of O atom shifted to 531.86, 532.27, and 532.74 eV at pH 5.1, and shifted to 531.97, 532.48, and 532.91 eV at pH 6.1; the peaks at 399.66 and 401.79 eV of N atom shifted to 399.99, 401.87 eV at pH 5.1, and to 400.18, 401.91 eV at pH 6.1. These shift can be explained that the electronic density around O and N atoms decreased when coordinated with Cu(II) [36], both O and N were involved for Cu(II) coordination. The XPS characteristics for β-zeolite and EDA@β-zeolite before and after Cu(II) adsorption were listed in Table 4 and Table 5. The XPS spectral characteristics provided the evidence that the oxygen atoms of β-zeolite formed coordination bonds with Cu(II), and nitrogen atoms on EDA@β-zeolite played significant roles in Cu(II) coordination [39].

### 3.6. Regeneration

The reusability of a sorbent is essential to evaluate its performance and application basing on the economic consideration. As the results presented in pH effect part, few Cu(II) was adsorbed at very low pH, implying that acid treatment was a possible regeneration method for the Cu(II)-loaded adsorbents. The regeneration of β-zeolite and EDA@β-zeolite was performed in 0.05 mol/L HCl solution. As shown in Figure 10, after four cycles of the adsorption and desorption process, no significant decrease in Cu(II) adsorption was observed, the removal capacity for Cu(II) was more than 95%. The results indicated that the Cu(II)-loaded EDA@β-zeolite composite could be efficiently regenerated by 0.05 mol/L HCl and reused without obvious decrease in Cu(II) removal capability. The proposed EDA@β-zeolite composite in present work possesses potential long-term use with low replacement cost, and is a promising candidate for the industrial removal of Cu(II) from considerable volume of aqueous solution.

### 3.7. Effect of Coexisting Ions

For industrial application, the effect of the presence of Fe(III) and Ca(II) on the performance in Cu(II) removal is studied. Figure 11 illustrates the competitive adsorption of Cu(II) onto β-zeolite and EDA@β-zeolite in mixed multi-metal solution. As shown in Figure 11, adsorption capacity follows such trend: Fe(III) > Cu(II) > Ca(II). It means that if Fe(III) is present in a real solution, β-zeolite and EDA@β-zeolite will also absorb Fe(III).

## 4. Conclusions

The ethylenediamine functionalized β-zeolite (EDA@β-zeolite) was prepared by a post-grafting method, and its removal behaviors toward Cu(II) from aqueous solution were investigated by batch techniques. The removal ability was significantly enhanced after ethylenediamine modification, and the removal was strongly dependent on pH, the surface complexation dominated the removal process. The kinetic process of Cu(II) removal could be well described by the pseudo-second-order kinetic model. Langmuir model fitted the sorption isotherms better than Freundlich and D–R models. High temperature was favorable for Cu(II) removal, thermodynamic data suggested that the removal process of Cu(II) on β-zeolite and EDA@β-zeolite were spontaneous and endothermic. The XPS results showed the functional groups of EDA@β-zeolite containing N elements were involved in removal process by forming complexes with Cu(II). The EDA@β-zeolite composite can be reused at least four times without decreasing removal capacity. The results in present work provide an effective candidate for the removal of Cu(II) from considerable volume of contaminated water around industrial facilities.

## Figures and Tables

**Figure 1 molecules-26-00978-f001:**
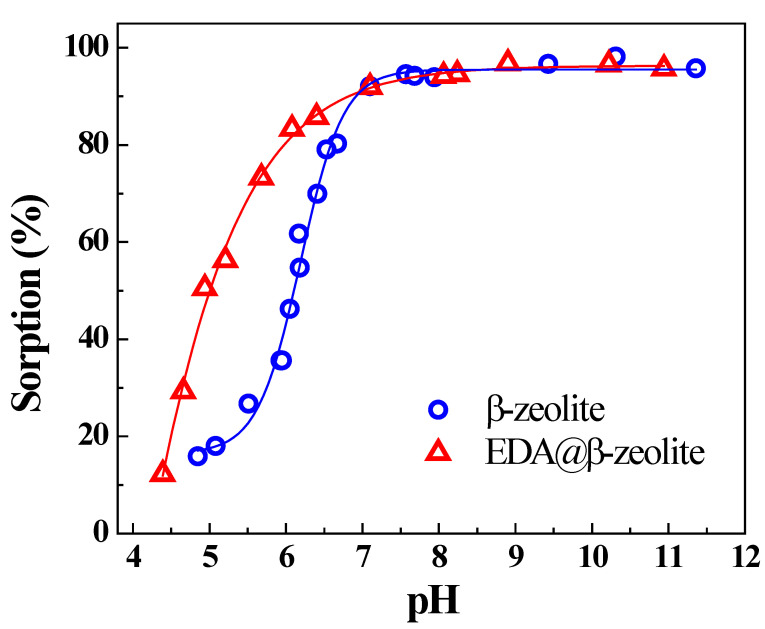
Effect of pH on Cu(II) adsorption onto β-zeolite and EDA@β-zeolite. *m/V* = 0.75 g/L, *I* = 0.01 mol/L NaCl, *T* = 25 °C, [Cu(II)] = 1.00 × 10^−4^ mol/L.

**Figure 2 molecules-26-00978-f002:**
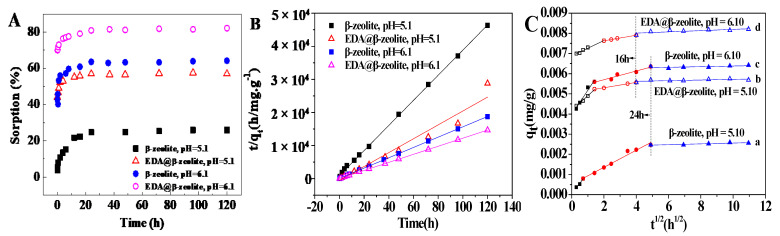
The kinetics of Cu(II) adsorption onto β-zeolite and EDA@β-zeolite. (**A**) Effect of contact time; (**B**) Pseudo-second-order adsorption kinetics plot; (**C**) Weber-Morris adsorption kinetics plot. *m/V* = 0.75 g/L, *I* = 0.01 mol/L NaCl, *T* = 25 °C, [Cu(II)] = 1.00 × 10^−4^ mol/L.

**Figure 3 molecules-26-00978-f003:**
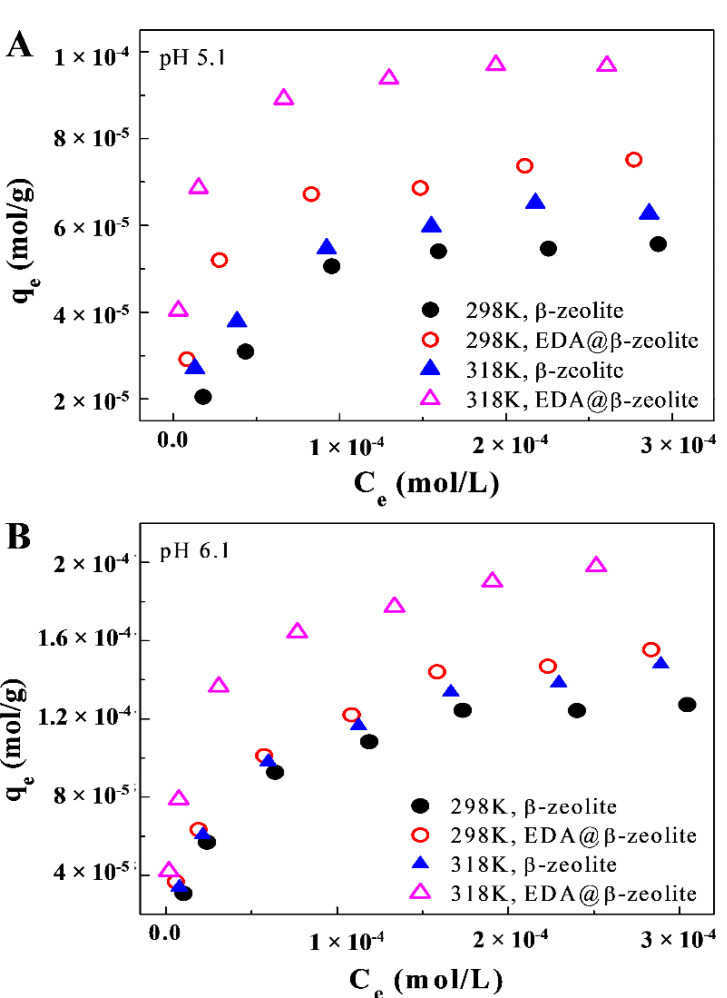
Effect of temperature on Cu(II) adsorption onto β-zeolite and EDA@β-zeolite at pH = 5.10 (**A**) and pH = 6.10 (**B**). *m/V* = 0.75 g/L, *I* = 0.01 mol/L NaCl, [Cu(II)] = 1.00 × 10^−4^ mol/L.

**Figure 4 molecules-26-00978-f004:**
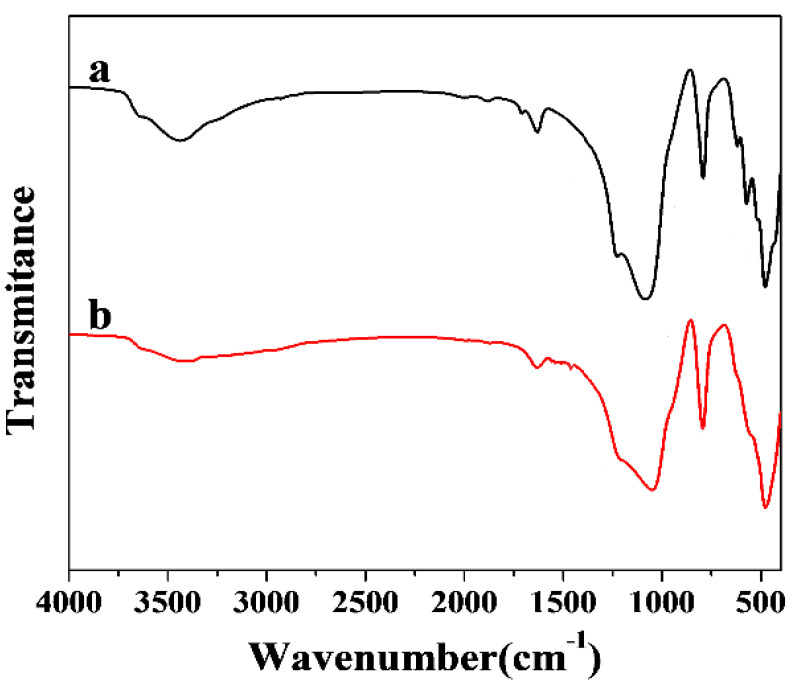
FT-IR spectra of (>a) β-zeolite and (b) EDA@β-zeolite.

**Figure 5 molecules-26-00978-f005:**
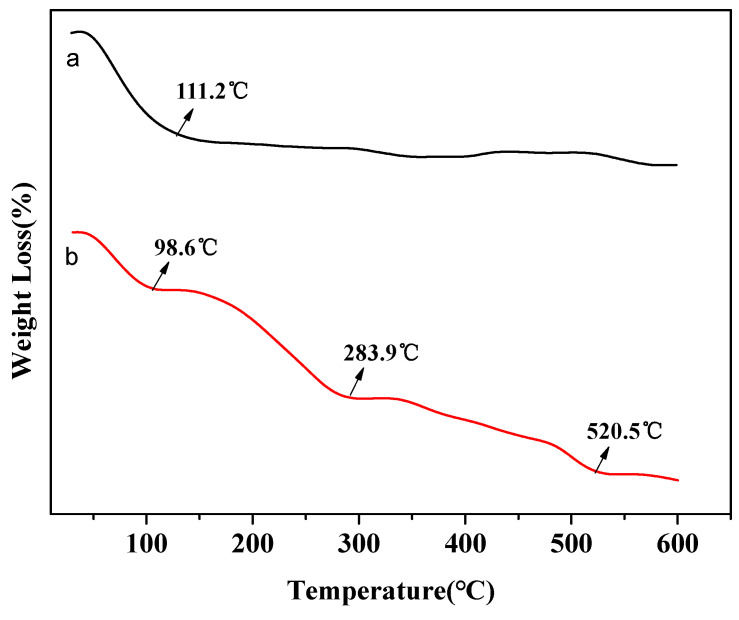
Thermal curves of (a) β-zeolite and (b) EDA@β-zeolite [15].

**Figure 6 molecules-26-00978-f006:**
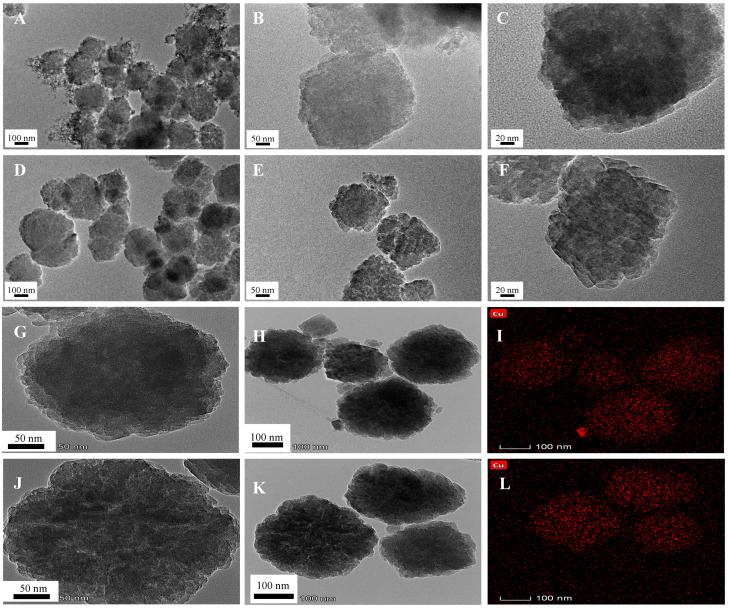
(**A**–**C**) TEM images of β-zeolite; (**D**–**F**) TEM images of EDA@β-zeolite; (**G**,**H**) TEM images of Cu(II)-loaded β-zeolite, pH = 6.10; (**I**) EDX Mapping of Cu(II)-loaded β-zeolite, pH = 6.10; (**J**,**K**) TEM images of Cu(II)-loaded EDA@β-zeolite, pH = 6.10; (**L**) EDX Mapping of Cu(II)-loaded EDA@β-zeolite, pH = 6.10.

**Figure 7 molecules-26-00978-f007:**
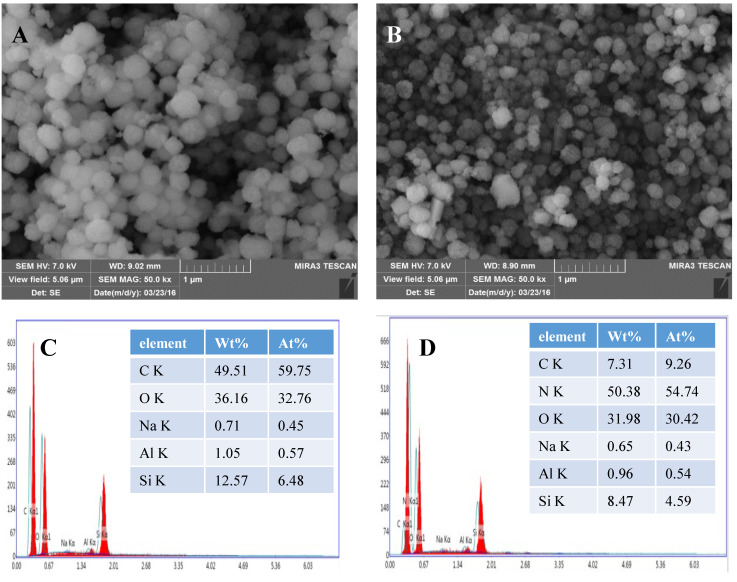
(**A**) SEM images of β-zeolite; (**B**) SEM images of EDA@β-zeolite; (**C**) EDS analysis of β-zeolite; (**D**) EDS analysis of EDA@β-zeolite.

**Figure 8 molecules-26-00978-f008:**
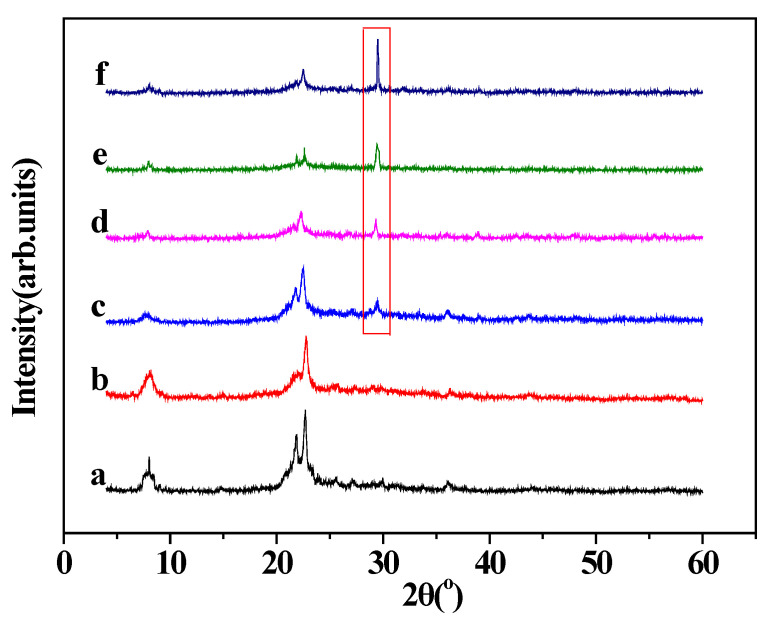
XRD patterns of (a) β-zeolite; (b) EDA@β-zeolite; (c) β-zeolite after Cu(II) adsorption, pH = 5.10; (d) β-zeolite after Cu(II) adsorption, pH = 6.10; (e) EDA@β-zeolite after Cu(II) adsorption, pH = 5.10; (f) EDA@β-zeolite after Cu(II) adsorption, pH = 6.10.

**Figure 9 molecules-26-00978-f009:**
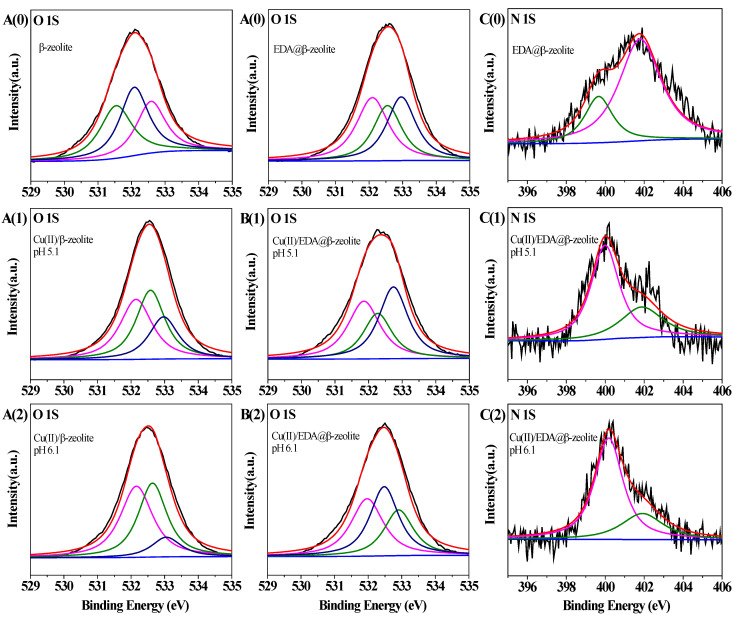
O 1s spectra of β-zeolite before Cu(II) adsorption [**A**(0)] and after Cu(II) adsorption [**A**(1),**A**(2)]. O 1s and N 1s spectra of EDA@β-zeolite before Cu(II) adsorption [**B**(0),**C**(0)] and after Cu(II) adsorption [**B**(1), **C**(1), **B**(2), and **C**(2)]. For **A**(1), **B**(1), and **C**(1), pH = 5.10; for **A**(2), **B**(2), and **C**(2), pH = 6.10.

**Figure 10 molecules-26-00978-f010:**
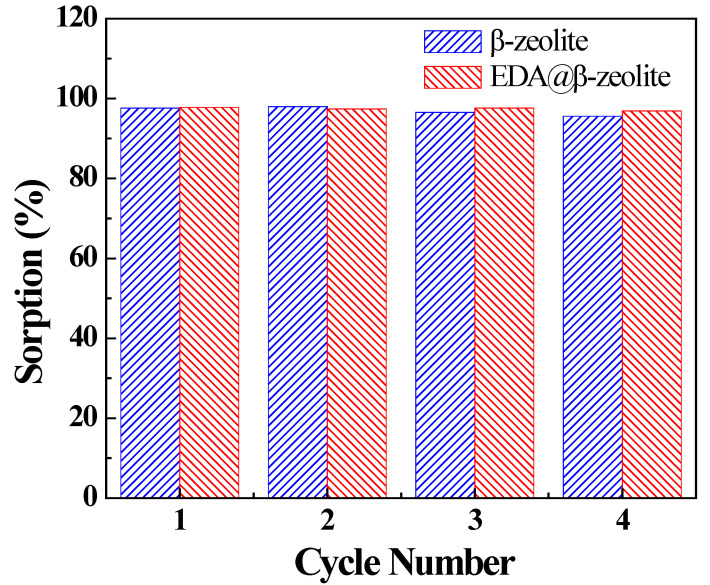
Recycling of β-zeolite and EDA@β-zeolite in Cu(II) removal from aqueous solution. *m/V* = 0.75 g/L, *I* = 0.01 mol/L NaCl, *T* = 25 °C, [Cu(II)] = 1.00 × 10^−4^ mol/L.

**Figure 11 molecules-26-00978-f011:**
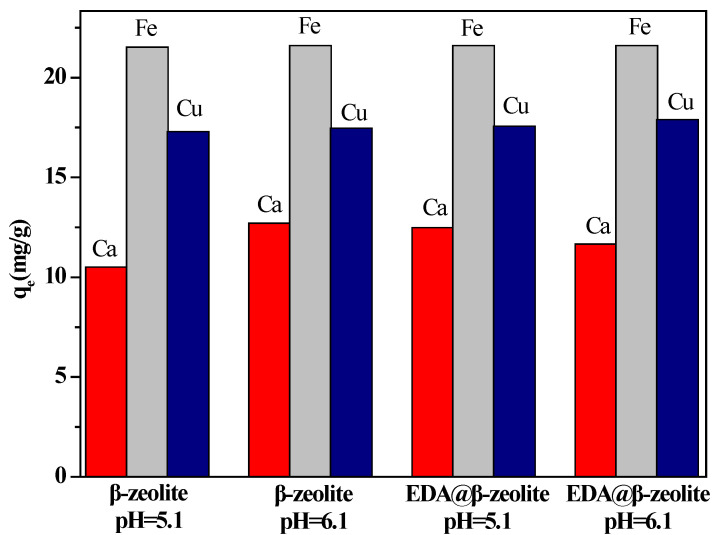
Effect of Fe(III) and Ca(II) in Cu(II) removal from aqueous solution. *m/V* = 0.75 g/L, *I* = 0.01 mol/L NaCl, *T* = 25 °C, [Ca(II)] = 1.00 × 10^−4^ mol/L, [Fe(III)] = 1.00 × 10^−4^ mol/L, [Cu(II)] = 1.00 × 10^−4^ mol/L.

**Table 1 molecules-26-00978-t001:** Kinetic parameters of pseudo-first order and pseudo-second order model.

pH	Sample	*q_e_* (mg/g)	Pseudo-First Order		Pseudo-Second Order	
			*q*_*e*1_(mg/g)	R_1_^2^	*q*_*e*2_(mg/g)	R_2_^2^
5.10	β-zeolite	0.00272	0.001939	0.8259	0.002636	0.9992
5.10	EDA@β-zeolite	0.00454	0.000694	0.8711	0.004515	0.9999
6.10	β-zeolite	0.00645	0.001462	0.7267	0.006412	0.9999
6.10	EDA@β-zeolite	0.00821	0.000726	0.8992	0.008203	0.9999

**Table 2 molecules-26-00978-t002:** Kinetic parameters of Weber–Morris model.

pH	Sample	Parameters		R_1_^2^
5.10	β-zeolite	K_a1_ = 0.0010	C_a1_ = 0.0001	0.9432
		K_a2_ = 0.0004	C_a2_ = 0.0005	0.9748
		K_a3_ = 2 × 10^−5^	C_a3_ = 0.0024	0.7126
5.10	EDA@β-zeolite	K_b1_ = 0.0007	C_b1_ = 0.0042	0.9893
		K_b2_ = 0.0001	C_b2_ = 0.0050	0.9958
		K_b3_ = 2 × 10^−5^	C_b3_ = 0.0056	0.4200
6.10	β-zeolite	K_c1_ = 0.0012	C_c1_ = 0.0040	0.9605
		K_c2_ = 0.0002	C_c2_ = 0.0053	0.9378
		K_c3_ = 1 × 10^−5^	C_c3_ = 0.0062	0.5071
6.10	EDA@β-zeolite	K_d1_ = 0.0004	C_d1_ = 0.0069	0.9786
		K_d2_ = 0.0001	C_d2_ = 0.0074	0.9980
		K_d3_ = 3 × 10^−5^	C_d3_ = 0.0079	0.6320

**Table 3 molecules-26-00978-t003:** The linear fit of ln *K_d_* vs. *C_e_* and thermodynamic parameters for Cu(II) adsorption onto β-zeolite and EDA@β-zeolite.

pH	*T* (K)	Sample	Ln *K_d_* = A *C_e_* + B			Thermodynamic Data		
			A	B	R	Δ*G*^0^ (kJ/mol)	Δ*H*^0^ (kJ/mol)	Δ*S*^0^ (J/mol·K)
5.10	298	β-zeolite	−2194.71	2.92	0.9924	−1.0716	0.1994	4.2649
5.10	298	EDA@β-zeolite	−3944.03	3.39	0.9465	−1.2208	0.3622	5.3123
5.10	318	β-zeolite	−3233.53	3.18	0.9501	−1.1569	0.1994	
5.10	318	EDA@β-zeolite	−5381.72	3.77	0.9122	−1.3271	0.3357	
6.10	298	β-zeolite	−1885.44	3.21	0.9850	−1.1663	0.3338	5.0339
6.10	298	EDA@β-zeolite	−3418.45	3.58	0.9272	−1.2754	0.4147	5.6715
6.10	318	β-zeolite	−4138.52	3.55	0.9782	−1.2670	0.3086	
6.10	318	EDA@β-zeolite	−5032.34	4.01	0.9091	−1.3888	0.3864	

**Table 4 molecules-26-00978-t004:** XPS results for β-zeolite before and after Cu(II) adsorption.

Element	β-Zeolite			Cu(II)-Loaded β-Zeolite (pH = 5.10)			Cu(II)-Loaded β-Zeolite (pH = 6.10)			Assignments
	BE (eV)	FWHM	Area (%)	BE (eV)	FWHM	Area (%)	BE (eV)	FWHM	Area (%)	
O 1S	532.10	1.14	37.26	532.15	1.14	37.75	532.16	1.14	46.42	Si-O-Al
	532.55	1.01	28.83	532.58	1.01	38.37	532.62	1.01	42.43	-OH
	533.01	1.03	33.91	533.05	1.03	42.14	533.07	1.03	11.15	water

**Table 5 molecules-26-00978-t005:** XPS results for EDA@β-zeolite before and after Cu(II) adsorption.

Element	EDA@β-zeolite			Cu(II)-Loaded EDA@β-Zeolite (pH = 5.10)			Cu(II)-Loaded EDA@β-Zeolite (pH = 6.10)			Assignments
	BE(eV)	FWHM	Area(%)	BE(eV)	FWHM	Area(%)	BE(eV)	FWHM	Area(%)	
O 1S	531.54	1.14	32.94	531.86	1.14	35.47	531.97	1.14	35.88	Si-O-Al
	532.05	1.01	37.86	532.27	1.01	24.72	532.48	1.01	38.31	-OH
	532.58	1.03	29.20	532.74	1.03	39.81	532.91	1.03	25.81	water
N 1S	399.66	1.72	23.42	399.99	1.72	67.45	400.18	1.72	72.35	-NH
	401.79	2.56	76.58	401.87	2.56	32.55	401.91	2.56	27.65	-NH_2_

## Data Availability

Not applicable.

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
