# Peer review of "Removal of Cu(II) Contamination from Aqueous Solution by Ethylenediamine@β-Zeolite Composite"

_molecules, 2021, doi:10.3390/molecules26040978_

Round 1
Reviewer 1 Report
I am satisfied with the changes. Still, one expression may be incorrectly interpreted by the reader. By writing: "The remaining NaCl was further washed with water and methanol. ", one can interpret, that you wash crystals of NaCl with water and methanol, but what you mean is: "NaCl remaining in the sample was further washed with water and methanol. ", or something similar. It is only a suggestion for the authors. For me, the article could be suitable for publication.
Author Response
Many thanks for your comments and suggestions of our manuscript. We have revised the expression as you suggested in line 78.
Reviewer 2 Report
Authors addressed all the comments.
Author Response
Thank you for your time and comments.
This manuscript is a resubmission of an earlier submission. The following is a list of the peer review reports and author responses from that submission.
Round 1
Reviewer 1 Report
The subject of the paper is important since the removal of copper and other heavy metals from ground water and other water objects is a kin problem. In principle, the methodology used by the authors allowed to arrive at the valuable conclusions, however the papers suffers from a few flaws and the following comments should be taken into account:
- I failed to find the EDA loading on the zeolite. Also, the loading variation should be studied.
- The new format (template) is responsible for the bad look of 4-digit figures in Tables (Table 4). Maybe, this table can be moved to SI?
- The proposed material is supposed to be used for purification of real water objects, therefore, the effect of the presence of Fe(3+) and Ca(2+) ions on the performance in copper removal should be studied.
Author Response
Response to Reviewer 1 Comments 1. I failed to find the EDA loading on the zeolite. Also, the loading variation should be studied. Response 1: The FT-IR and TGA of zeolite and EDA@zeolite is added to the manuscript, the differences between zeolite and EDA@zeolite are also explained in detail in line 271. 2. The new format (template) is responsible for the bad look of 4-digit figures in Tables (Table 4). Maybe, this table can be moved to SI? Response 2: Thank you for your suggestion. Table 4 is moved to SI 2. 3. The proposed material is supposed to be used for purification of real water objects, therefore, the effect of the presence of Fe(3+) and Ca(2+) ions on the performance in copper removal should be studied. Response 3: The presence of Fe(3+) and Ca(2+) ions in copper removal is studied. The result is stated in the manuscript in detail in line 363.Reviewer 2 Report
The authors present the results from the research concerning cleaning the water from copper ions. The use zeolite beta modified with ethylenediamine. Removal of Cu(ii) ions has been increased upon organic modification of zeolite. The authors focus on removal kinetics. Langmuir model is found to have the best fitting to this kinetics. TEM and SEM pictures, XPS spectroscopy were also covered. The article is written in good English, is well constructed and clearly presented. I recommed the article for publishing in ‘Molecules’ after small changes.
- Minor editing revision is required (see line 42, lines 49-53: different font, Table 4 – fonts)
- Table 4 is very hard to read – a reader cannot distinguish between the values in different columns
- Line 63: „with the aid of spectroscopy” – specify what kind of spectrocopy
- 1 line 75-76: why the solid was washed with NaCl solution? Is this a correct statement: „The remaining NaCl was further washed with water and methanol.”
Author Response
Response to Reviewer 2 Comments
- Minor editing revision is required (see line 42, lines 49-53: different font, Table 4 – fonts)
Response 1: The mistakes of editing are examined and revised.
- Table 4 is very hard to read – a reader cannot distinguish between the values in different columns
Response 2: Table 4 is moved to SI, and some significant digits are deduced.
- Line 63: „with the aid of spectroscopy” – specify what kind of spectroscopy
Response 3: The relevant characterization methods are listed in the manuscript in line 66, such as X-ray powder diffraction (XRD), X-ray photoelectron spectroscopy (XPS), etc.
- 1 line 75-76: why the solid was washed with NaCl solution? Is this a correct statement: „The remaining NaCl was further washed with water and methanol.”
Response 4: By investigation we found that EDA can react with water produces a by-product. To ensure the purity of the product, the solid was washed with NaCl solution, then the remaining NaCl was washed with water and methanol.
Reviewer 3 Report
The manuscript by Liu et al. titled “Removal of Cu(II) Contamination from Aqueous Solution by Ethylenediamine@β-Zeolite Composite” submitted to Molecules is describing the method of preparation of EDA modified Beta zeolite followed by use of this material for sorption of Cu(II). I would like to address following comments:
- For fig. 6 is stated: a new peak at 29.3◦ appears after Cu(II) adsorption.
How this peak was assigned?
- How the TEM of zeolite after adsorption would like? Please, provide data and comment.
- “A novel ethylenediamine functionalized β-zeolite (EDA@β-zeolite)” is this really novel? Was published in 2017 paper (ref.15 and earlier).
- I am missing in the paper the description of the interaction of EDA with zeolite. How the functionalization works? This is crucial for description of the system. Would be good to add this in introduction (around line 57).
- Leaching tests of EDA for modified beta zeolites is required. How stable is the modified EDA@beta?
- Figure 2. A – I am missing the comparison of modified and non-modified zeolites in pH higher than 8. What was the motivation to use pH of 5.1 and 6.1, why not higher? – please add comment in the paper.
- What was the composition of EDA@Beta? How much of EDA was in the modified zeolite? TG profile or chemical analysi is required.
- TEM images Fig 4 A and E are very unclear – should be removed.
Technical issues:
Table 4 is not very clear, please, modify it to show data clearly.
Figure 4. Scalebars in TEM images are not visible at all.
The language corrections should be performed, e.g.:
Line 50 – heavy mentals ???
Line 64 - This composite is potential in the successful…
line 93 - , calculated Cu(II) stock solution, - what does it mean?
Line 116 – “literatures”
Line 266 – “nitrogen content of EDA@β-zeolite was presented for 54.74%.”
… and more.
Overall, I recommend the major revision of this manuscript before publication.
Author Response
Response to Reviewer 3 Comments
- For fig. 6 is stated: a new peak at 29.3◦ appears after Cu(II) adsorption. How this peak was assigned?
Response 1: By comparing PDF-2 standard cards, the peak was assigned to CuCl2.
- How the TEM of zeolite after adsorption would like? Please, provide data and comment.
Response 2: The TEM of zeolite and EDA@β-zeolite after adsorption are added in line 289.
- “A novel ethylenediamine functionalized β-zeolite (EDA@β-zeolite)” is this really novel? Was published in 2017 paper (ref.15 and earlier).
Response 3: Yeah, the material has been published earlier, the statement is not accurate. I have changed it.
- I am missing in the paper the description of the interaction of EDA with zeolite. How the functionalization works? This is crucial for description of the system. Would be good to add this in introduction (around line 57).
Response 4: The description of the interaction of EDA with zeolite is added to the manuscript in line 58: one amine ligand of ethylenediamine is used as organic linkers to coordinate zeolite, the other amine ligand is applied to adsorb mental ions.
- Leaching tests of EDA for modified beta zeolites is required. How stable is the modified EDA@beta?
Response 5: Leaching tests of EDA for modified beta zeolites are added in the SI 1. The results indicated that EDA@β-zeolite has good stability at pH ranges from 4 to 12.
- Figure 2. A – I am missing the comparison of modified and non-modified zeolites in pH higher than 8. What was the motivation to use pH of 5.1 and 6.1, why not higher? – please add comment in the paper.
Response 6: Due to the formation of insoluble substance in high pH, the influence of contact time was explored at pH = 5.1 and 6.1.
- What was the composition of EDA@Beta? How much of EDA was in the modified zeolite? TG profile or chemical analysi is required.
Response 7: FT-IR and TGA is added in the manuscript in line 271to explain the composition of EDA@β-zeolite.
- TEM images Fig 4 A and E are very unclear – should be removed.
Response 8: TEM images Fig 4 A and E are removed. The TEM images has been re-labeled in line 289.
Technical issues:
Table 4 is not very clear, please, modify it to show data clearly.
Response: Table 4 is moved to SI, and some significant digits are deduced.
Figure 4. Scalebars in TEM images are not visible at all.
Response: New scalebars are written on TEM images in line 289.
The language corrections should be performed, e.g.:
Line 50 – heavy mentals ???
Line 64 - This composite is potential in the successful…
line 93 - , calculated Cu(II) stock solution, - what does it mean?
Line 116 – “literatures”
Line 266 – “nitrogen content of EDA@β-zeolite was presented for 54.74%.”
Response: The language corrections listed above have been performed as follows:
“heavy mentals” corrected to “mental ions”
“This composite is potential in the successful…” corrected to “This composite is a potential material in the successful…”
“calculated Cu(II) stock solution” corrected to “Cu(II) solution”
“literatures” corrected to “references”
“nitrogen content of EDA@β-zeolite was presented for 54.74%.” corrected to “It is worth noting that the N mass percentage of β-zeolite and EDA@β-zeolite was about 0 and 54.74%, respectively.”
Round 2
Reviewer 1 Report
The authors took into account all the comments and revised the manuscript.
Reviewer 3 Report
I was writing about the need of language corrections and they were not performed. Still there are things like: "mental ions." - line 60 and many others.